# Capsules with Inverted Dot-Product Attention Routing

**Yao-Hung Hubert Tsai**[1,2]**, Nitish Srivastava**[1]**, Hanlin Goh**[1]**, Ruslan Salakhutdinov**[1,2]
[1]Apple Inc., [2]Carnegie Mellon University
`yaohungt@cs.cmu.edu`,`{nitish_srivastava,hanlin,rsalakhutdinov}@apple.com`

## Abstract

We introduce a new routing algorithm for capsule networks, in which a child capsule is routed to a parent based only on agreement between the parent's state and the child's vote. The new mechanism 1) designs routing via inverted dot-product attention; 2) imposes Layer Normalization as normalization; and 3) replaces sequential iterative routing with concurrent iterative routing. When compared to previously proposed routing algorithms, our method improves performance on benchmark datasets such as CIFAR-10 and CIFAR-100, and it performs at-par with a powerful CNN (ResNet-18) with $4\times$ fewer parameters. On a different task of recognizing digits from over-layed digit images, the proposed capsule model performs favorably against CNNs given the same number of layers and neurons per layer. We believe that our work raises the possibility of applying capsule networks to complex real-world tasks. Our code is publicly available at: `https://github.com/apple/ml-capsules-inverted-attention-routing`. An alternative implementation is available at: `https://github.com/yaohungt/Capsules-Inverted-Attention-Routing/blob/master/README.md`.

## 1 Introduction

Capsule Networks (CapsNets) represent visual features using groups of neurons. Each group (called a "capsule") encodes a feature and represents one visual entity. Grouping all the information about one entity into one computational unit makes it easy to incorporate priors such as "a part can belong to only one whole" by routing the entire part capsule to its parent whole capsule. Routing is mutually exclusive among parents, which ensures that one part cannot belong to multiple parents. Therefore, capsule routing has the potential to produce an interpretable hierarchical parsing of a visual scene. Such a structure is hard to impose in a typical convolutional neural network (CNN). This hierarchical relationship modeling has spurred a lot of interest in designing capsules and their routing algorithms (Sabour et al., 2017; Hinton et al., 2018; Wang & Liu, 2018; Zhang et al., 2018; Li et al., 2018; Rajasegaran et al., 2019; Kosiorek et al., 2019).

In order to do routing, each lower-level capsule votes for the state of each higher-level capsule. The higher-level (parent) capsule aggregates the votes, updates its state, and uses the updated state to explain each lower-level capsule. The ones that are well-explained end up routing more towards that parent. This process is repeated, with the vote aggregation step taking into account the extent to which a part is routed to that parent. Therefore, the states of the hidden units and the routing probabilities are inferred in an iterative way, analogous to the M-step and E-step, respectively, of an Expectation-Maximization (EM) algorithm. Dynamic Routing (Sabour et al., 2017) and EM-routing (Hinton et al., 2018) can both be seen as variants of this scheme that share the basic iterative structure but differ in terms of details, such as their capsule design, how the votes are aggregated, and whether a non-linearity is used.

We introduce a novel routing algorithm, which we called *Inverted Dot-Product Attention Routing*. In our method, the routing procedure resembles an inverted attention mechanism, where dot products are used to measure agreement. Specifically, the higher-level (parent) units compete for the attention of the lower-level (child) units, instead of the other way around, which is commonly used in attention

Figure 1: Illustration of a Capsule network with a backbone block, 3 convolutional capsule layers, 2 fully-connected capsule layers, and a classifier. The first convolutional capsule layer is called the primary capsule layer. The last fully-connected capsule layer is called the class capsule layer.

models. Hence, the routing probability directly depends on the agreement between the parent's pose (from the previous iteration step) and the child's vote for the parent's pose (in the current iteration step). We also propose two modifications for our routing procedure – (1) using Layer Normalization (Ba et al., 2016) as normalization, and (2) doing inference of the latent capsule states and routing probabilities jointly across multiple capsule layers (instead of doing it layer-wise). These modifications help scale up the model to more challenging datasets.

Our model achieves comparable performance as the state-of-the-art convolutional neural networks (CNNs), but with much fewer parameters, on CIFAR-10 (95.14% test accuracy) and CIFAR-100 (78.02% test accuracy). We also introduce a challenging task to recognize single and multiple overlapping objects simultaneously. To be more precise, we construct the DiverseMultiMNIST dataset that contains both single-digit and overlapping-digits images. With the same number of layers and the same number of neurons per layer, the proposed CapsNet has better convergence than a baseline CNN. Overall, we argue that with the proposed routing mechanism, it is no longer impractical to apply CapsNets on real-world tasks. We will release the source code to reproduce the experiments.

## 2 CAPSULE NETWORK ARCHITECTURE

An example of our proposed architecture is shown in Figure 1. The backbone is a standard feed-forward convolutional neural network. The features extracted from this network are fed through another convolutional layer. At each spatial location, groups of 16 channels are made to create capsules (we assume a 16-dimensional pose in a capsule). LayerNorm is then applied across the 16 channels to obtain the primary capsules. This is followed by two convolutional capsule layers, and then by two fully-connected capsule layers. In the last capsule layer, each capsule corresponds to a class. These capsules are then used to compute logits that feed into a softmax to computed the classification probabilities. Inference in this network requires a feed-forward pass up to the primary capsules. After this, our proposed routing mechanism (discussed in the next section) takes over.

In prior work, each capsule has a pose and some way of representing an activation probability. In Dynamic Routing CapsNets (Sabour et al., 2017), the pose is represented by a vector and the activation probability is implicitly represented by the norm of the pose. In EM Routing CapsNets (Hinton et al., 2018), the pose is represented by a matrix and the activation probability is determined by the EM algorithm. In our work, we consider a matrix-structured pose in a capsule. We denote the capsules in layer $L$ as $\mathbf{P}^L$ and the $i$-th capsule in layer $L$ as $\mathbf{p}_i^L$. The pose $\mathbf{p}_i^L \in \mathbb{R}^{d_L}$ in a vector form and will be reshaped to $\mathbb{R}^{\sqrt{d_L} \times \sqrt{d_L}}$ when representing it as a matrix, where $d_L$ is the number of hidden units grouped together to make capsules in layer $L$. The activation probability is not explicitly represented. By doing this, we are essentially asking the network to represent the absence of a capsule by some special value of its pose.

## 3 INVERTED DOT-PRODUCT ATTENTION ROUTING

The proposed routing process consists of two steps. The first step computes the agreement between lower-level capsules and higher-level capsules. The second step updates the pose of the higher-level capsules.

**Step 1: Computing Agreement:** To determine how capsule $j$ in layer $L + 1$ ($\mathbf{p}_j^{L+1}$) agrees with capsule $i$ in layer $L$ ($\mathbf{p}_i^L$), we first transform the pose $\mathbf{p}_i^L$ to the **vote** $\mathbf{v}_{ij}^L$ for the pose $\mathbf{p}_j^{L+1}$. This

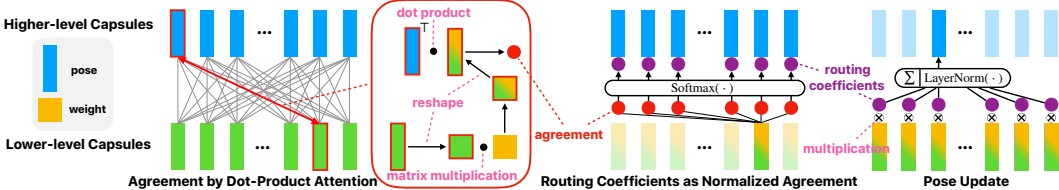

Figure 2: Illustration of the Inverted Dot-Product Attention Routing with the pose admitting matrix structure.

---

**Procedure 1** Inverted Dot-product Attention Routing algorithm returns updated **poses** of the capsules in layer $L + 1$ given **poses** in layer $L$ and $L + 1$ and **weights** between layer $L$ and $L + 1$.

1: **procedure** INVERTED DOT-PRODUCT ATTENTION ROUTING($\mathbf{P}^L, \mathbf{P}^{L+1}, \mathbf{W}^L$)
2:     for all capsule $i$ in layer $L$ and capsule $j$ in layer $(L + 1)$: $\mathbf{v}_{ij}^L \leftarrow \mathbf{W}_{ij}^L \cdot \mathbf{p}_i^L$     ▷ vote
3:     for all capsule $i$ in layer $L$ and capsule $j$ in layer $(L + 1)$: $a_{ij}^L \leftarrow {\mathbf{p}_j^{L+1}}^\top \cdot \mathbf{v}_{ij}^L$  ▷ agreement
4:     for all capsule $i$ in layer $L$: $r_{ij}^L \leftarrow \exp(a_{ij}^L) / \sum_{j'} \exp(a_{ij'}^L)$     ▷ routing coefficient
5:     for all capsule $j$ in layer $(L + 1)$: $\mathbf{p}_j^{L+1} \leftarrow \sum_i r_{ij}^L \mathbf{v}_{ij}^L$     ▷ pose update
6:     for all capsule $j$ in layer $(L + 1)$: $\mathbf{p}_j^{L+1} \leftarrow \texttt{LayerNorm}(\mathbf{p}_j^{L+1})$     ▷ normalization
7:     **return** $\mathbf{P}^{L+1}$

---

transformation is done using a learned **transformation matrix** $\mathbf{W}_{ij}^L$:

$$\mathbf{v}_{ij}^L = \mathbf{W}_{ij}^L \cdot \mathbf{p}_i^L, \tag{1}$$

where the matrix $\mathbf{W}_{ij}^L \in \mathbb{R}^{d_{L+1} \times d_L}$ if the pose has a vector structure and $\mathbf{W}_{ij}^L \in \mathbb{R}^{\sqrt{d_{L+1}} \times \sqrt{d_L}}$ (requires $d_{L+1} = d_L$) if the pose has a matrix structure. Next, the **agreement** ($a_{ij}^L$) is computed by the dot-product similarity between a pose $\mathbf{p}_j^{L+1}$ and a vote $\mathbf{v}_{ij}^L$:

$$a_{ij}^L = {\mathbf{p}_j^{L+1}}^\top \cdot \mathbf{v}_{ij}^L. \tag{2}$$

The pose $\mathbf{p}_j^{L+1}$ is obtained from the previous iteration of this procedure, and will be set to $\mathbf{0}$ initially.

**Step 2: Computing Poses:** The agreement scores $a_{ij}^L$ are passed through a softmax function to determine the routing probabilities $r_{ij}^L$:

$$r_{ij}^L = \frac{\exp(a_{ij}^L)}{\sum_{j'} \exp(a_{ij'}^L)}, \tag{3}$$

where $r_{ij}^L$ is an inverted attention score representing how higher-level capsules compete for attention of lower-level capsules. Using the routing probabilities, we **update the pose** $\mathbf{p}_j^{L+1}$ for capsule $j$ in layer $L + 1$ from all capsules in layer $L$:

$$\mathbf{p}_j^{L+1} = \text{LayerNorm}\left(\sum_i r_{ij}^L \mathbf{v}_{ij}^L\right). \tag{4}$$

We adopt Layer Normalization (Ba et al., 2016) as the **normalization**, which we empirically find it to be able to improve the convergence for routing. The routing algorithm is summarized in Procedure 1 and Figure 2.

## 4 INFERENCE AND LEARNING

To explain how inference and learning are performed, we use Figure 1 as an example. Note that the choice of the backbone, the number of capsules layers, the number of capsules per layer, the design of the classifier may vary for different sets of experiments. We leave the discussions of configurations in Sections 5 and 6, and in the Appendix.

---

**Procedure 2** Inference. Inference returns **class logits** given **input images** and **parameters** for the model. Capsule layer 1 denotes the primary capsules layer and layer $N$ denotes the class capsules layer.

---

1: **procedure** INFERENCE($\mathbf{I}; \boldsymbol{\theta}, \mathbf{W}^{1:N-1}$)
    */* Pre-Capsules Layers: backbone features extraction */*
2:     $\mathbf{F} \leftarrow \text{backbone}(\mathbf{I}; \boldsymbol{\theta})$                ▷ backbone feature
    */* Capsules Layers: initialization */*
3:     $\mathbf{P}^1 \leftarrow \text{LayerNorm}(\text{convolution}(\mathbf{F}; \boldsymbol{\theta}))$          ▷ primary capsules
4:     for $L$ in layers 2 to $N$: $\mathbf{P}^L \leftarrow \mathbf{0}s$          ▷ non-primary capsules
    */* Capsules Layers (1st Iteration): sequential routing */*
5:     **for** $L$ in layers 1 to $(N-1)$ **do**
6:         $\mathbf{P}^{L+1} \leftarrow \text{Routing}(\mathbf{P}^L, \mathbf{P}^{L+1}; \mathbf{W}^L)$       ▷ non-primary capsules
    */* Capsules Layers (2nd to tth Iteration): concurrent routing */*
7:     **for** $(t-1)$ iterations **do**
8:         for $L$ in layers 1 to $(N-1)$: $\bar{\mathbf{P}}^{L+1} \leftarrow \text{Routing}(\mathbf{P}^L, \mathbf{P}^{L+1}; \mathbf{W}^L)$
9:         for $L$ in layers 2 to $N$: $\mathbf{P}^L \leftarrow \bar{\mathbf{P}}^L$       ▷ non-primary capsules
    */* Post-Capsules Layers: classification */*
10:     for all capsule $i$ in layer $N$: $\hat{y}_i \leftarrow \text{classifier}(\mathbf{p}_i^N; \boldsymbol{\theta})$      ▷ class logits
11:     **return** $\hat{\mathbf{y}}$

---

## 4.1 INFERENCE

For ease of exposition, we decompose a CapsNet into pre-capsule, capsule and post-capsule layers.

**Pre-Capsule Layers:** The goal is to obtain a **backbone feature** $\mathbf{F}$ from the input image $\mathbf{I}$. The backbone model can be either a single convolutional layer or ResNet computational blocks (He et al., 2016).

**Capsule Layers:** The **primary capsules** $\mathbf{P}^1$ are computed by applying a convolution layer and Layer Normalization to the backbone feature $\mathbf{F}$. The **non-primary capsules** layers $\mathbf{P}^{2:N}$ are initialized to be zeros [1]. For the first iteration, we perform one step of routing sequentially in each capsule layer. In other words, the primary capsules are used to update their parent convolutional capsules, which are then used to update the next higher-level capsule layer, and so on. After doing this first pass, the rest of the routing iterations are performed concurrently. Specifically, all capsule layers look at their preceding lower-level capsule layer and perform one step of routing simultaneously. This procedure is an example of a parallel-in-time inference method. We call it "concurrent routing" as it concurrently performs routing between capsules layers per iteration, leading to better parallelism. Figure 3 illustrates this procedure from routing iteration 2 to $t$. It is worth noting that, our proposed variant of CapsNet is a weight-tied concurrent routing architecture with Layer Normalization, which Bai et al. (2019) empirically showed could converge to fixed points.

Previous CapsNets (Sabour et al., 2017; Hinton et al., 2018) used sequential layer-wise iterative routing between the capsules layers. For example, the model first performs routing between layer $L-1$ and layer $L$ for a few iterations. Next, the model performs routing between layer $L$ and $L+1$ for a few iterations. When unrolled, this sequential iterative routing defines a very deep computational graph with a single path going from the inputs to the outputs. This deep graph could lead to a vanishing gradients problem and limit the depth of a CapsNet that can be trained well, especially if any squashing non-linearities are present. With concurrent routing, the training can be made more stable, since each iteration has a more cumulative effect.

**Post-Capsule Layers:** The goal is to obtain the **predicted class logits** $\hat{\mathbf{y}}$ from the last capsule layer (the class capsules) $\mathbf{P}^N$. In our CapsNet, we use a linear classifier for class $i$ in class capsules: $\hat{y}_i = \text{classifier}(\mathbf{p}_i^N)$. This classifier is shared across all the class capsules.

---

[1] As compared to 0 initialization, we observe that a random initialization leads to similar converged performance but slower convergence speed. We also tried to learn biases for capsules' initialization, which results in similar converged performance and same convergence speed. As a summary, we initialize the capsule's value to 0 for simplicity.

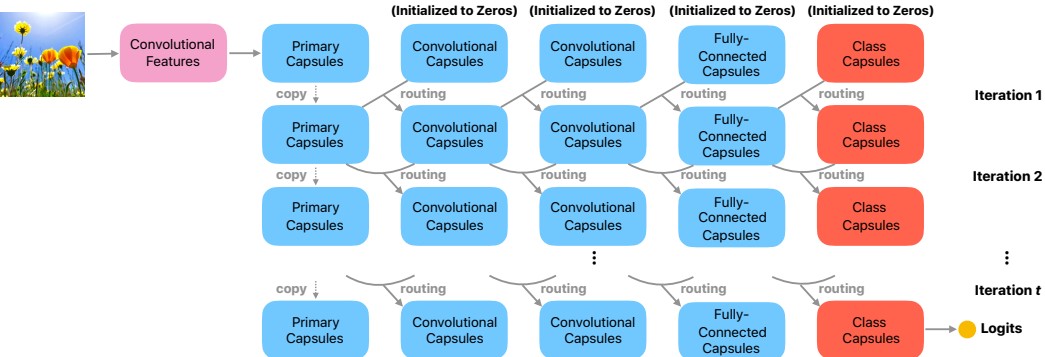

Figure 3: Illustration of the proposed concurrent routing from iteration 2 to $t$ with the example in Figure 1. The concurrent routing is a parallel-in-time routing procedure for all capsules layers.

Table 1: Classification results on CIFAR-10/CIFAR-100 without ensembling models. We report the best performance for CapsNets when considering 1 to 5 routing iterations. We report the performance from the best test model for baseline routing methods, our routing method, and ResNet (He et al., 2016).

| Method | Backbone | Test Accuracy (# of parameters) | |
|---|---|---|---|
| | | CIFAR-10 | CIFAR-100 |
| Dynamic Routing (Sabour et al., 2017) | simple | 84.08% (7.99M) | 56.96% (31.59M) |
| EM Routing (Hinton et al., 2018) | simple | 82.19 (0.45M) | 37.73% (0.50M) |
| Inverted Dot-Product Attention Routing (ours) | simple | 85.17 (0.56M) | 57.32% (1.46M) |
| Dynamic Routing (Sabour et al., 2017) | ResNet | 92.65% (12.45M) | 71.70% (36.04M) |
| EM Routing (Hinton et al., 2018) | ResNet | 92.15% (1.71M) | 58.08% (1.76M) |
| Inverted Dot-Product Attention Routing (ours) | ResNet | 95.14% (1.83M) | 78.02% (2.80M) |
| Baseline CNN (simple) | | 87.10% (18.92M) | 62.30% (19.01M) |
| ResNet-18 (He et al., 2016) | | 95.11% (11.17M) | 77.92% (11.22M) |

## 4.2 LEARNING

We update the parameters $\theta, \mathbf{W}^{1:N-1}$ by stochastic gradient descent. For multiclass classification, we use multiclass cross-entropy loss. For multilabel classification, we use binary cross-entropy loss. We also tried Margin loss and Spread loss which are introduced by prior work (Sabour et al., 2017; Hinton et al., 2018). However, these losses do not give us better performance against cross-entropy and binary cross-entropy losses.

## 4.3 COMPARISONS WITH EXISTING CAPSNET MODELS

Having described our model in detail, we can now place the model in the context of previous work. In the following table, we list the major differences among different variants of CapsNets.

| | Dynamic Routing (Sabour et al., 2017) | EM Routing (Hinton et al., 2018) | Inverted Dot-Product Attention Routing (ours) |
|---|---|---|---|
| Routing | sequential iterative routing | sequential iterative routing | concurrent iterative routing |
| Poses | vector | matrix | matrix |
| Activations | n/a (norm of poses) | determined by EM | n/a |
| Non-linearity | Squash function | n/a | n/a |
| Normalization | n/a | n/a | Layer Normalization |
| Loss Function | Margin loss | Spread loss | Cross Entropy/Binary Cross Entropy |

## 5 EXPERIMENTS ON CIFAR-10 AND CIFAR-100

CIFAR-10 and CIFAR-100 datasets (Krizhevsky et al., 2009) consist of small $32 \times 32$ real-world color images with $50,000$ for training and $10,000$ for evaluation. CIFAR-10 has 10 classes, and CIFAR-100 has 100 classes. We choose these natural image datasets to demonstrate our method since they correspond to a more complex data distribution than digit images.

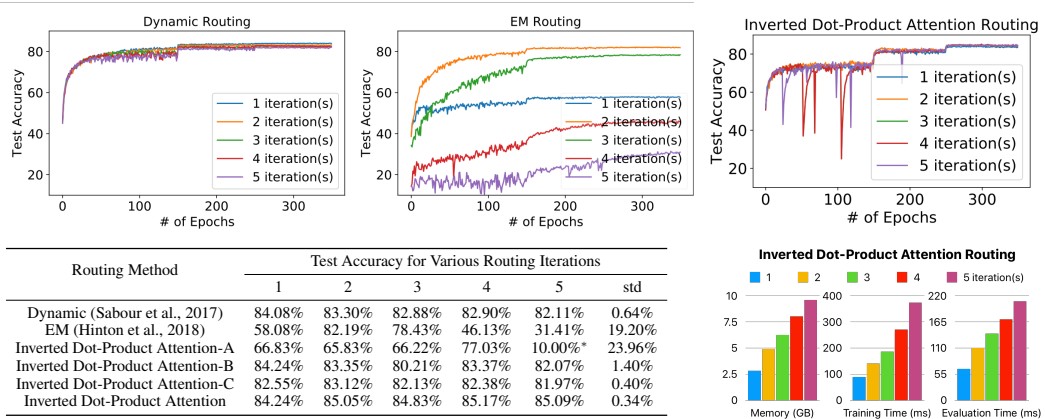

| Routing Method | Test Accuracy for Various Routing Iterations | | | | | |
| --- | --- | --- | --- | --- | --- | --- |
| | 1 | 2 | 3 | 4 | 5 | std |
| Dynamic (Sabour et al., 2017) | 84.08% | 83.30% | 82.88% | 82.90% | 82.11% | 0.64% |
| EM (Hinton et al., 2018) | 58.08% | 82.19% | 78.43% | 46.13% | 31.41% | 19.20% |
| Inverted Dot-Product Attention-A | 66.83% | 65.83% | 66.22% | 77.03% | 10.00%* | 23.96% |
| Inverted Dot-Product Attention-B | 84.24% | 83.35% | 80.21% | 83.37% | 82.07% | 1.40% |
| Inverted Dot-Product Attention-C | 82.55% | 83.12% | 82.13% | 82.38% | 81.97% | 0.40% |
| Inverted Dot-Product Attention | 84.24% | 85.05% | 84.83% | 85.17% | 85.09% | 0.34% |

Figure 4: Convergence analysis for CapsNets on CIFAR-10 with simple backbone model. **Top**: convergence plots for different routing mechanisms. **Bottom left**: classification results with respect to different routing iterations. Inverted Dot-Product Attention-A denotes our routing approach without Layer Normalization. Inverted Dot-Product Attention-B denotes our routing approach with sequential routing. Inverted Dot-Product Attention-C denotes our routing approach with activations in capsules. * indicates a uniform prediction. **Bottom right**: memory usage and inference time for the proposed Inverted Dot-Product Attention Routing. For fairness, the numbers are benchmarked using the same 8-GPU machine with batch size 128. Note that for future comparisons, we refer the readers to an alternative implementation for our model: `https://github.com/yaohungt/Capsules-Inverted-Attention-Routing/blob/master/README.md`. It uses much less memory, has a significantly faster inference speed, and retains the same performance.

**Comparisons with other CapsNets and CNNs:** In Table 1, we report the test accuracy obtained by our model, along with other CapsNets and CNNs. Two prior CapsNets are chosen: Dynamic Routing CapsNets (Sabour et al., 2017) and EM Routing CapsNets (Hinton et al., 2018). For each CapsNet, we apply two backbone feature models: simple convolution followed by ReLU nonlinear activation and a ResNet (He et al., 2016) backbone. For CNNs, we consider a baseline CNN with 3 convolutional layers followed by 1 fully-connected classifier layer. ResNet-18 is selected as a representative of SOTA CNNs. See Appendix A.1 for detailed configurations.

First, we compare previous routing approaches against ours. In a general trend, the proposed CapsNets perform better than the Dynamic Routing CapsNets, and the Dynamic Routing CapsNets perform better than EM Routing CapsNets. The performance differs more on CIFAR-100 than on CIFAR-10. For example, with simple convolutional backbone, EM Routing CapsNet can only achieve 37.73% test accuracy while ours can achieve 57.32%. Additionally, for all CapsNets, we see improved performance when replacing a single convolutional backbone with ResNet backbone. This result is not surprising since ResNet structure has better generalizability than a single convolutional layer. For the number of parameters, ours and EM Routing CapsNets have much fewer as compared to Dynamic Routing CapsNets. The reason is due to different structures of capsule's pose. Ours and EM Routing CapsNets have matrix-structure poses, and Dynamic Routing CapsNets have vector-structure poses. With matrix structure, weights between capsules are only $O(d)$ with $d$ being pose's dimension; with vector structure, weights are $O(d^2)$. To conclude, combining the proposed Inverted Dot-Product Attention Routing with ResNet backbone gives us both the advantages of a low number of parameters and high performance.

Second, we discuss the performance difference between CNNs and CapsNets. We see that, with a simple backbone (a single convolutional layer), it is hard for CapsNets to reach the same performance as CNNs. For instance, our routing approach can only achieve 57.32% test accuracy on CIFAR-100 while the baseline CNN achieves 62.30%. However, with a SOTA backbone structure (ResNet backbone), the proposed routing approach can reach competitive performance (95.14% on CIFAR-10) as compared to the SOTA CNN model (ResNet-18 with 95.11% on CIFAR-10).

**Convergence Analysis:** In Figure 4, top row, we analyze the convergence for CapsNets with respect to the number of routing iterations. The optimization hyperparameters are chosen optimally for each routing mechanism. For Dynamic Routing CapsNets (Sabour et al., 2017), we observe a mild performance drop when the number of iterations increases. For EM Routing CapsNets (Hinton et al., 2018), the best-performed number of iterations is 2. Increasing or decreasing this number severely hurts the performance. For our proposed routing mechanism, we find a positive correlation between performance and number of routing iterations. The performance variance is also the smallest among

| Method | Pose Structure | Test Acc. | # params. |
|--------|----------------|-----------|-----------|
| CapsNet* | vector | 83.39% | 42.48M |
| CapsNet | matrix | 80.59% | 9.96M |
| CapsNet | vector | 85.74% | 42.48M |
| Baseline CNN | | 79.81% | 19.55M |

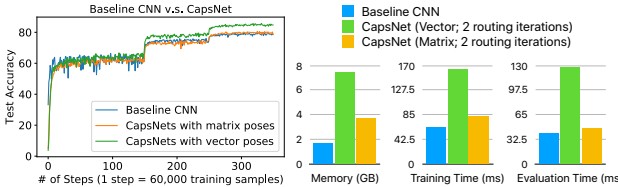

Figure 5: Table and convergence plot for baseline CNN and CapsNets with different pose structures. We consider the same optimizer, the same number of layers, and the same number of neurons per layer for the models. For fairness, memory usage/ inference time are benchmarked using the same 8-GPU machine with batch size 128. CapsNet* denotes the Dynamic routing method and CapsNet denotes our proposed Inverted Dot-Product Attention Routing method. Note that for future comparisons, we refer the readers to an alternative implementation for our model: `https://github.com/yaohungt/Capsules-Inverted-Attention-Routing/blob/master/README.md`. It uses much less memory, has a significantly faster inference speed, and retains the same performance. The memory usage and inference time in this implementation now are **only marginally** higher than the baseline CNN.

the three routing mechanisms. This result suggests our approach has better optimization and stable inference. However, selecting a larger iteration number may not be ideal since memory usage and inference time will also increase (shown in the bottom right in Figure 4). Note that, we observe sharp performance jitters during training when the model has not converged (especially when the number of iterations is high). This phenomenon is due to applying LayerNorm on a low-dimensional vector. The jittering is reduced when we increase the pose dimension in capsules.

**Ablation Study:** Furthermore, we inspect our routing approach with the following ablations: 1) Inverted Dot-Product Attention-A: without Layer Normalization; 2) Inverted Dot-Product Attention-B: replacing concurrent to sequential iterative routing; and 3) Inverted Dot-Product Attention-C: adding activations in capsules [2]. The results are presented in Figure 4 bottom row. When removing Layer Normalization, performance dramatically drops from our routing mechanism. Notably, the prediction becomes uniform when the iteration number increases to 5. This result implies that the normalization step is crucial to the stability of our method. When replacing concurrent with sequential iterative routing, the positive correlation between performance and iteration number no longer exists. This fact happens in the Dynamic Routing CapsNet as well, which also uses sequential iterative routing. When adding activations to our capsule design, we obtain a performance deterioration. Typically, squashing activations such as sigmoids make it harder for gradients to flow, which might explain this. Discovering the best strategy to incorporate activations in capsule networks is an interesting direction for future work.

# 6 EXPERIMENTS ON DIVERSEMULTIMNIST

The goal in this section is to compare CapsNets and CNNs when they have the same number of layers and the same number of neurons per layer. Specifically, we would like to examine the difference of the representation power between the routing mechanism (in CapsNets) and the pooling operation (in CNNs). A challenging setting is considered in which objects may be overlapping with each other, and there may be a diverse number of objects in the image. To this end, we construct the DiverseMultiMNIST dataset which is extended from MNIST (LeCun et al., 1998), and it contains both single-digit and two overlapping digit images. The task will be multilabel classification, where the prediction is said to be correct if and only if the recognized digits match all the digits in the image. We plot the convergence curve when the model is trained on $21M$ images from DiverseMultiMNIST. Please see Appendix B.2 for more details on the dataset and Appendix B.1 for detailed model configurations. The results are reported in Figure 5.

First, we compare our routing method against the Dynamic routing one. We observe an improved performance from the CapsNet* to the CapsNet (83.39% to 85.74% with vector-structured poses). The result suggests a better viewpoint generalization for our routing mechanism.

Second, we compare baseline CNN against our CapsNet. From the table, we see that CapsNet has better test accuracy compared to CNN. For example, the CapsNet with vector-structured poses reaches 85.74% test accuracy, and the baseline CNN reaches 79.81% test accuracy. In our CNN implementation, we use average pooling from the last convolutional layer to its next fully-connected

---

[2]We consider the same kind of capsules activations as described in EM Routing CapsNets (Hinton et al., 2018).

layer. We can see that having a routing mechanism works better than pooling. However, one may argue that the pooling operations requires no extra parameter but routing mechanism does, and hence it may not be fair to compare their performance. To address this issue, in the baseline CNN, we replace the pooling operation with a fully-connected operation. To be more precise, instead of using average pooling, we learn the entire transformation matrix from the last convolutional layer to its next fully-connected layer. This procedure can be regarded as considering pooling with learnable parameters. After doing this, the number of parameters in CNN increases to $42.49M$, and the corresponding test accuracy is $84.84\%$, which is still lower than $85.74\%$ from the CapsNet. We conclude that, when recognizing overlapping and diverse number of objects, the routing mechanism has better representation power against the pooling operation.

Last, we compare CapsNet with different pose structures. The CapsNet with vector-structured poses works better than the CapsNet with matrix-structured poses ($80.59\%$ vs $85.74\%$). However, the former requires more parameters, more memory usage, and more inference time. If we increase the number of parameters in the matrix-pose CapsNet to $42M$, its test accuracy rises to $91.17\%$. Nevertheless, the model now requires more memory usage and inference time as compared to using vector-structured poses. We conclude that more performance can be extracted from vector-structured poses but at the cost of high memory usage and inference time.

## 7   RELATED WORK

The idea of grouping a set of neurons into a capsule was first proposed in Transforming Auto-Encoders (Hinton et al., 2011). The capsule represented the multi-scale recognized fragments of the input images. Given the transformation matrix, Transforming Auto-Encoders learned to discover capsules' instantiation parameters from an affine-transformed image pair. Sabour et al. (2017) extended this idea to learn part-whole relationships in images systematically. Hinton et al. (2018) cast the routing mechanism as fitting a mixture of Gaussians. The model demonstrated an impressive ability for recognizing objects from novel viewpoints. Recently, Stacked Capsule Auto-Encoders (Kosiorek et al., 2019) proposed to segment and compose the image fragments without any supervision. The work achieved SOTA results on unsupervised classification. However, despite showing promising applications by leveraging inherent structures in images, the current literature on capsule networks has only been applied on datasets of limited complexity. Our proposed new routing mechanism instead attempts to apply capsule networks to more complex data.

Our model also relates to Transformers (Vaswani et al., 2017) and Set Transformers (Lee et al., 2019), where dot-product attention is also used. In the language of capsules, a Set Transformer can be seen as a model in which a higher-level unit can choose to pay attention to $K$ lower-level units (using $K$ attention heads). Our model inverts the attention direction (lower-level units "attend" to parents), enforces exclusivity among routing to parents and does not impose any limits on how many lower-level units can be routed to any parent. Therefore, it combines the ease and parallelism of dot-product routing derived from a Transformer, with the interpretability of building a hierarchical parsing of a scene derived from capsule networks.

There are other works presenting different routing mechanisms for capsules. Wang & Liu (2018) formulated the Dynamic routing (Sabour et al., 2017) as an optimization problem consisting of a clustering loss and a KL regularization term. Zhang et al. (2018) generalized the routing method within the framework of weighted kernel density estimation. Li et al. (2018) approximated the routing process with two branches and minimized the distributions between capsules layers by an optimal transport divergence constraint. Phaye et al. (2018) replaced standard convolutional structures before capsules layers by densely connected convolutions. It is worth noting that this work was the first to combine SOTA CNN backbones with capsules layers. Rajasegaran et al. (2019) proposed DeepCaps by stacking $10+$ capsules layers. It achieved $92.74\%$ test accuracy on CIFAR-10, which was the previous best for capsule networks. Instead of looking for agreement between capsules layers, Choi et al. (2019) proposed to learn deterministic attention scores only from lower-level capsules. Nevertheless, without agreement, their best-performed model achieved only $88.61\%$ test accuracy on CIFAR-10. In contrast to these prior work, we present a combination of inverted dot-product attention routing, layer normalization, and concurrent routing. To the best of our knowledge, we are the first to show that capsule networks can achieve comparable performance against SOTA CNNs. In particular, we achieve $95.14\%$ test accuracy for CIFAR-10 and $78.02\%$ for CIFAR-100.

# 8 CONCLUSION AND FUTURE WORK

In this work, we propose a novel Inverted Dot-Product Attention Routing algorithm for Capsule networks. Our method directly determines the routing probability by the agreements between parent and child capsules. Routing algorithms from prior work require child capsules to be explained by parent capsules. By removing this constraint, we are able to achieve competitive performance against SOTA CNN architectures on CIFAR-10 and CIFAR-100 with the use of a low number of parameters. We believe that it is no longer impractical to apply capsule networks to datasets with complex data distribution. Two future directions can be extended from this paper:

- In the experiments, we show how capsules layers can be combined with SOTA CNN backbones. The optimal combinations between SOTA CNN structures and capsules layers may be the key to scale up to a much larger dataset such as ImageNet.

- The proposed concurrent routing is as a parallel-in-time and weight-tied inference process. The strong connection with Deep Equilibrium Models (Bai et al., 2019) can potentially lead us to infinite-iteration routing.

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

## A   MODEL CONFIGURATIONS FOR CIFAR-10/CIFAR-100

### A.1   MODEL SPECIFICATIONS

The configuration choices of Dynamic Routing CapsNets and EM Routing CapsNets are followed by prior work (Sabour et al., 2017; Hinton et al., 2018). We empirically find their configurations perform the best for their routing mechanisms (instead of applying our network configurations to their routing mechanisms). The optimizers are chosen to reach the best performance for all models. We list the model specifications in Table 2, 3, 4, 5, 6, 7, 8, and 9.

We only show the specifications for CapsNets with a simple convolutional backbone. When considering a ResNet backbone, two modifications are performed. First, we replace the simple feature backbone with ResNet feature backbone. Then, the input dimension of the weights after the backbone is set as 128. A ResNet backbone contains a $3 \times 3$ convolutional layer (output 64-dim.), three 64-dim. residual building block (He et al., 2016) with stride 1, and four 128-dim. residual building block with stride 2. The ResNet backbone returns a $16 \times 16 \times 128$ tensor.

For the optimizers, we use stochastic gradient descent with learning rate 0.1 for our proposed method, baseline CNN, and ResNet-18 (He et al., 2016). We use Adam (Kingma & Ba, 2014) with learning rate 0.001 for Dynamic Routing CapsNets and Adam with learning rate 0.01 for EM Routing CapsNets. We decrease the learning rate by 10 times when the model trained on 150 epochs and 250 epochs, and there are 350 epochs in total.

### A.2   DATA AUGMENTATIONS

We consider the same data augmentation for all networks. During training, we first pad four zero-value pixels to each image and randomly crop the image to the size $32 \times 32$. Then, we horizontally flip the image with probability 0.5. During evaluation, we do not perform data augmentation. All the model is trained on a 8-GPU machine with batch size 128.

## B   MODEL CONFIGURATIONS FOR DIVERSE_MULTIMNIST

### B.1   MODEL SPECIFICATIONS

To fairly compare CNNs and CapsNets, we fix the number of layers and the number of neurons per layer in the models. These models consider the design: 36x36 image → 18x18x1024 neurons → 8x8x1024 neurons → 6x6x1024 neurons → 640 neurons → 10 class logits. The configurations are presented in Table 10, 11, and 12. We also fix the optimizers across all the models. We use stochastic gradient descent with learning rate 0.1 and decay the learning rate by 10 times when the models trained on 150 steps and 250 steps. One step corresponds to $60,000$ training samples, and we train the models with a total of 350 steps.

## B.2 DATASET CONSTRUCTION

Diverse_MultiMNIST contains both single-digit and overlapping-digit images. We generate images on the fly and plot the test accuracy for training models over $21M$ ($21M = 350(\text{steps}) \times 60,000(\text{images})$) generated images. We also generate the test images, and for each evaluation step, there are $10,000$ test images. Note that we make sure the training and the test images are from the disjoint set. In the following, we shall present how we generate the images. We set the probability of generating a single-digit image as $\frac{1}{6}$ and the probability of generating an overlapping-digit image as $\frac{5}{6}$.

The single-digit image in DiverseMultiMNIST training/ test set is generated by shifting digits in MNIST (LeCun et al., 1998) training/ test set. Each digit is shifted up to $4$ pixels in each direction and results in $36 \times 36$ image.

Following Sabour et al. (2017), we generate overlapping-digit images in DiverseMultiMNIST training/ test set by overlaying two digits from the same training/ test set of MNIST. Two digits are selected from different classes. Before overlaying the digits, we shift the digits in the same way which we shift for the digit in a single-digit image. After overlapping, the generated image has size $36 \times 36$.

We consider no data augmentation for both training and evaluation. All the model is trained on a 8-GPU machine with batch size $128$.

Table 2: Baseline CNN for CIFAR-10.

| Operation | Output Size |
|---|---|
| input_dim=3, output_dim=1024, 3x3 conv, stride=2, padding=1
ReLU | 16x16x1024 |
| input_dim=1024, output_dim=1024, 3x3 conv, stride=2, padding=1
ReLU + Batch Norm | 8x8x1024 |
| 2x2 average pooling, padding=0 | 4x4x1024 |
| input_dim=1024, output_dim=1024, 3x3 conv, stride=2, padding=1
ReLU + Batch Norm | 2x2x1024 |
| 2x2 average pooling, padding=0 | 1x1x1024 |
| Flatten | 1024 |
| input_dim=1024, output_dim=10, linear | 10 |

Table 3: Baseline CNN for CIFAR-100.

| Operation | Output Size |
|---|---|
| input_dim=3, output_dim=1024, 3x3 conv, stride=2, padding=1
ReLU | 16x16x1024 |
| input_dim=1024, output_dim=1024, 3x3 conv, stride=2, padding=1
ReLU + Batch Norm | 8x8x1024 |
| 2x2 average pooling, padding=0 | 4x4x1024 |
| input_dim=1024, output_dim=1024, 3x3 conv, stride=2, padding=1
ReLU + Batch Norm | 2x2x1024 |
| 2x2 average pooling, padding=0 | 1x1x1024 |
| Flatten | 1024 |
| input_dim=1024, output_dim=100, linear | 100 |

Table 4: Dynamic Routing with Simple Backbone for CIFAR-10.

| Operation | Output Size |
|---|---|
| input_dim=3, output_dim=256, 9x9 conv, stride=1, padding=0 ReLU | 24x24x256 |
| input_dim=256, output_dim=256, 9x9 conv, stride=2, padding=0 | 8x8x256 |
| Capsules reshape Squash | 8x8x32x8 |
| Capsules flatten | 2048x8 |
| Linear Dynamic Routing to 10 16-dim. capsules Squash | 10x16 |

Table 5: Dynamic Routing with Simple Backbone for CIFAR-100.

| Operation | Output Size |
|---|---|
| input_dim=3, output_dim=256, 9x9 conv, stride=1, padding=0 ReLU | 24x24x256 |
| input_dim=256, output_dim=256, 9x9 conv, stride=2, padding=0 | 8x8x256 |
| Capsules reshape Squash | 8x8x32x8 |
| Capsules flatten | 2048x8 |
| Linear Dynamic Routing to 100 16-dim. capsules Squash | 100x16 |

Table 6: EM Routing with Simple Backbone for CIFAR-10.

| Operation | Output Size |
|---|---|
| input_dim=3, output_dim=256, 4x4 conv, stride=2, padding=1 Batch Norm + ReLU | 16x16x256 |
| input_dim=256, output_dim=512, 1x1 conv, stride=1, padding=0 & input_dim=256, output_dim=32, 1x1 conv, stride=1, padding=0 Sigmoid | 16x16x512 & 16x16x32 |
| Capsules reshape (only for poses) | 16x16x32x4x4 & 16x16x32 |
| Conv EM Routing to 32 4x4-dim. capsules, 3x3 conv, stride=2 | 7x7x32x4x4 & 7x7x32 |
| Conv EM Routing to 32 4x4-dim. capsules, 3x3 conv, stride=1 | 5x5x32x4x4 & 5x5x32 |
| Capsules flatten | 800x4x4 & 800 |
| Linear EM Routing to 10 4x4-dim. capsules | 10x4x4 & 10 |

Table 7: EM Routing with Simple Backbone for CIFAR-100.

| Operation | Output Size |
|---|---|
| input_dim=3, output_dim=256, 4x4 conv, stride=2, padding=1
Batch Norm + ReLU | 16x16x256 |
| input_dim=256, output_dim=512, 1x1 conv, stride=1, padding=0
&
input_dim=256, output_dim=32, 1x1 conv, stride=1, padding=0
Sigmoid | 16x16x512
&
16x16x32 |
| Capsules reshape (only for poses) | 16x16x32x4x4
&
16x16x32 |
| Conv EM Routing to 32 4x4-dim. capsules, 3x3 conv, stride=2 | 7x7x32x4x4
&
7x7x32 |
| Conv EM Routing to 32 4x4-dim. capsules, 3x3 conv, stride=1 | 5x5x32x4x4
&
5x5x32 |
| Capsules flatten | 800x4x4
&
800 |
| Linear EM Routing to 100 4x4-dim. capsules | 100x4x4
&
100 |

Table 8: Proposed Inverted Dot-Product Attention Routing with Simple Backbone for CIFAR-10.

| Operation | Output Size |
|---|---|
| input_dim=3, output_dim=256, 3x3 conv, stride=2, padding=1
ReLU | 16x16x256 |
| input_dim=256, output_dim=512, 1x1 conv, stride=1, padding=0
Layer Norm | 16x16x512 |
| Capsules reshape | 16x16x32x4x4 |
| Conv Inverted Dot-Product Attention Routing to 32 4x4-dim. capsules, 3x3 conv, stride=2
Layer Norm | 7x7x32x4x4 |
| Conv Inverted Dot-Product Attention Routing to 32 4x4-dim. capsules, 3x3 conv, stride=1
Layer Norm | 5x5x32x4x4 |
| Capsules flatten | 800x4x4 |
| Linear Inverted Dot-Product Attention Routing to 10 4x4-dim. capsules
Layer Norm | 10x4x4 |
| Reshape | 10x16 |
| input_dim=16, output_dim=1, linear | 10x1 |
| Reshape | 10 |

Table 9: Proposed Inverted Dot-Product Attention Routing with Simple Backbone for CIFAR-100.

| Operation | Output Size |
|---|---|
| input_dim=3, output_dim=128, 3x3 conv, stride=2, padding=1
ReLU | 16x16x128 |
| input_dim=128, output_dim=1152, 1x1 conv, stride=1, padding=0
Layer Norm | 16x16x1152 |
| Capsules reshape | 16x16x32x6x6 |
| Conv Inverted Dot-Product Attention Routing to 32 6x6-dim. capsules, 3x3 conv, stride=2
Layer Norm | 7x7x32x6x6 |
| Conv Inverted Dot-Product Attention Routing to 32 6x6-dim. capsules, 3x3 conv, stride=1
Layer Norm | 5x5x32x6x6 |
| Capsules flatten | 800x6x6 |
| Linear Inverted Dot-Product Attention Routing to 20 6x6-dim. capsules
Layer Norm | 20x6x6 |
| Linear Inverted Dot-Product Attention Routing to 100 6x6-dim. capsules
Layer Norm | 100x6x6 |
| Reshape | 100x36 |
| input_dim=36, output_dim=1, linear | 100x1 |
| Reshape | 100 |

Table 10: CapsNet with matrix-structured poses for DiverseMultiMNIST.

| Operation | Output Size |
|---|---|
| input_dim=3, output_dim=1024, 3x3 conv, stride=2, padding=1
ReLU | 18x18x1024 |
| input_dim=1024, output_dim=1024, 3x3 conv, stride=2, padding=0
Layer Norm | 8x8x1024 |
| Capsules reshape | 8x8x16x8x8 |
| Conv Inverted Dot-Product Attention Routing to 16 8x8-dim. capsules, 3x3 conv, stride=1
Layer Norm | 6x6x16x8x8 |
| Capsules flatten | 576x8x8 |
| Linear Inverted Dot-Product Attention Routing to 10 8x8-dim. capsules
Layer Norm | 10x8x8 |
| Reshape | 10x64 |
| input_dim=64, output_dim=1, linear
Sigmoid | 10x1 |
| Reshape | 10 |

Table 11: CapsNet with vector-structured poses for DiverseMultiMNIST.

| Operation | Output Size |
|---|---|
| input_dim=3, output_dim=1024, 3x3 conv, stride=2, padding=1 
 ReLU | 18x18x1024 |
| input_dim=1024, output_dim=1024, 3x3 conv, stride=2, padding=0 
 Layer Norm | 8x8x1024 |
| Capsules reshape | 8x8x16x64 |
| Conv Inverted Dot-Product Attention Routing to 16 64-dim. capsules, 3x3 conv, stride=1 
 Layer Norm | 6x6x16x64 |
| Capsules flatten | 576x64 |
| Linear Inverted Dot-Product Attention Routing to 10 8x8-dim. capsules 
 Layer Norm | 10x64 |
| input_dim=64, output_dim=1, linear 
 Sigmoid | 10x1 |
| Reshape | 10 |

Table 12: Baseline CNN for DiverseMultiMNIST.

| Operation | Output Size |
|---|---|
| input_dim=3, output_dim=1024, 3x3 conv, stride=2, padding=1 
 ReLU | 18x18x1024 |
| input_dim=1024, output_dim=1024, 3x3 conv, stride=2, padding=0 
 ReLU + Batch Norm | 8x8x1024 |
| input_dim=1024, output_dim=1024, 3x3 conv, stride=1, padding=0 
 ReLU + Batch Norm | 6x6x1024 |
| input_dim=1024, output_dim=640, linear | 6x6x640 |
| 6x6 average pooling, padding=0 | 1x1x640 |
| Flatten | 640 |
| input_dim=640, output_dim=10, linear 
 Sigmoid | 10 |

