# OpenReview forum: "Capsules with Inverted Dot-Product Attention Routing"
_ICLR.cc/2020/Conference — Accept (Poster)_

### Official Review · AnonReviewer1 · 2019-10-24
**Official Blind Review #1**

**Rating:** 6

**Review:**

In this paper, the authors propose a simple and effective routing algorithm for capsule networks. The paper is well written. A nice analysis of the proposed routing algorithm is provided. Experiments of varying the routing iterations demonstrate the stableability of proposed routing algorithm compared to others.

Here are some issues:
1. Would the authors release the code for reproducing the results in the paper? It will be helpful for future research in this area.

2. In Fig.5, it would be better to give some brief explanations about why CasNet (Matrix) occupies much more memory while possessing less parameters.

**Experience Assessment:**

I do not know much about this area.

**Review Assessment: Checking Correctness Of Derivations And Theory:**

I assessed the sensibility of the derivations and theory.

**Review Assessment: Checking Correctness Of Experiments:**

I assessed the sensibility of the experiments.

**Review Assessment: Thoroughness In Paper Reading:**

I read the paper at least twice and used my best judgement in assessing the paper.

---

> ### Author Response · Authors · 2019-11-14
> **Reponse to Review #1**
>
> We thank the Reviewer for their valuable feedback.
>
> [Code]
> We will release the code. Reproducibility is our priority.
>
> [Higher Memory Usage than CNNs]
> Two main reasons result in higher memory usage than CNNs.
>
> The first reason is that we perform iterative routing, which means that we perform routing multiple times. Since CapsNet is a weight-tied architecture, the memory usage scales linearly with the number of iterations. The details can be found in the illustration in Figure 3 and the bar plot of memory usage in Figure 4. Note that this phenomenon is also observed in Deep Equilibrium Models (DEQs) [1]. Inspired by DEQs, a potential solution is to refer to a fixed-point optimization for finding equilibrium points of the routing updates. Then, we can enjoy the benefits of constant memory.
>
> The second reason is that CapsNet uses the routing mechanism. As compared to the operations in CNNs, the routing mechanism performs agreement calculation between layers. This calculation introduces additional memory usage.However, we note that the routing mechanism is a dense operation, which means that we need to perform routing between all lower-layer capsules and all higher-layer capsules. We can instead randomly sample the capsules for routing, making the routing mechanism a sparse operation. We leave this dense-to-sparse modification as our future work.
>
>
> [1] “Deep equilibrium models”, S Bai, JZ Kolter, and V Koltun. NeurIPS 2019.

---

### Official Review · AnonReviewer5 · 2019-11-01
**Official Blind Review #5**

**Rating:** 8

**Review:**


This paper presents a new simpler routing mechanism for capsule networks and achieves good performance on real world data sets making use of this new capsule structure along with a restnet backbone. Strong performance on the cifar10 and cifar100 datasets are presented and the network outperforms earlier versions of capsule networks. This new structure also performs well on an augmented MNIST dataset of overlapping digits (similar to the one used by Sabour et al 2017).

Overall the paper is well written and presents solid results. The paper also presents a thorough comparison of two earlier versions of capsules which is a worthwhile contribution in its own right.

The paper could be improved by clearing up a few ambiguities:

- is the learning rate schedule the same for all three models? in figure 4 it looks like the learning rate is decayed at two distinct points for your model, but only one distinct point for both the EM and Dynamic routing models.

-"Notably, the prediction becomes random guess when the iteration number increases to 5." this sentence is a little confusing. Do you mean when the iteration number the performance is equivalent to not random assignments?

- This new algorithm requires that the capsules in L+1 have initialized poses with which to compare agreement between the poses in L. This is initial value seems like it may greatly effect the performance of the model. In the paper it is set to 0 and not expanded upon. It would be interesting to see if randomizing this value, or learning a bias for it would effect performance.

-unlike the two previous versions of capsules, the inverted dot product capsules show in figure 4 sudden huge decreases in test accuracy while training. These moments seem to be overcome quite quickly and the model ends up outperforming the other two. But it would be worth mentioning this behavior and perhaps attempting to explain it.


**Experience Assessment:**

I have published in this field for several years.

**Review Assessment: Checking Correctness Of Derivations And Theory:**

I assessed the sensibility of the derivations and theory.

**Review Assessment: Checking Correctness Of Experiments:**

I assessed the sensibility of the experiments.

**Review Assessment: Thoroughness In Paper Reading:**

I read the paper thoroughly.

---

> ### Author Response · Authors · 2019-11-14
> **Reponse to Review #5**
>
> We thank the Reviewer for the valuable feedback.
>
> [Learning Rate Scheduler]
> We use the same learning rate scheduler for all three models. The learning rate degrades by 0.1 on the 150th and 250th epochs. There are two distinct points for both the EM and Dynamic routing models, yet the second point is more noticeable when zooming in the convergence plot in Figure 4.
>
> The difference between all three models is the type of optimization method used. We use SGD for our model, and we use Adam for Dynamic and EM routing models. The type of the optimization method is selected to reach the best performance for each model. For example, SGD leads to worse performance than Adam for the Dynamic routing model.
>
> In the original submission, we have included these details in Section A.1 in Supplementary.
>
> [Uniform Prediction in Table in Figure 4]
> For Inverted Dot-Product Attention-A, when routing iteration increases to 5, we observe NaN values in neural network parameters. The prediction result becomes uniform across all classes. Since we consider a 10-class classification, the prediction accuracy becomes 10.00%. We rephrase the “random guess” to “uniform prediction” in the revised manuscript.
>
> [Non-zero Pose Initilization]
> We thank the Reviewer for raising the discussion about the capsule's initialization. As compared to 0 initialization, we observe that a random initialization leads to a similar converged performance but slower convergence speed. On the other hand, learning biases for capsules results in similar converged performance and same convergence speed. As a summary, we initialize the capsule's value to 0 for simplicity. We include the discussion in the revised manuscript.
>
> [Sudden Performance Jump in Convergence Plot in Figure 4]
> The phenomenon of the performance jump is due to applying LayerNorm on the low-dimensional pose. To be more precise, the dimension of the pose used in the convergence plot is 16, and we apply LayerNorm to these 16 units. When increasing the pose’s dimension, the jittering no longer existed. Nevertheless, we empirically find that it does not affect the model’s prediction result once the model converges.
>
> In the original submission, we have included these details in the last few sentences of the Convergence Analysis in Section 5.

---

### Official Review · AnonReviewer4 · 2019-11-02
**Official Blind Review #4**

**Rating:** 3

**Review:**

Authors improve upon dynamic routing between capsules by removing the squash function (norm normalization) and apply a layerNorm normalization instead. Furthermore, they experiment with concurrent routing rather than sequential routing (route all caps layers once, then all layers concurrently again and again). This is an interesting development since provides better gradient in conjunction with layerNorm. They report results on Cifar10 and Cifar100 and achieve similar to CNN (resnet) performance.

First, I want to point out that inverted attention is exactly what happens in dynamic routing (sabour et al 2017), proc. 1 line 4,5, and 7. In dynamic routing the dot product with the next layer capsule is calculated and then normalized over all next layer capsules. The only difference that I notice between alg. 1 here and proc. 1 there is replacement of squash with layer norm. There is no "reconstructing the layer bellow" in Dynamic routing as authors suggest in intro.

Second, the Capsules are promised to have better viewpoint generalizability than CNNs while having comparable performance. Replacing the 1 convolution layer with a ResNet backbone and replacing the activation with a classifier on top seems reducing the proposed CapsNet to the level of CNNs in terms of Viewpoint Generalization. Why should someone use this network rather than the ResNet itself? Fewer number of parameters by itself is not interesting, the reason it is reported usually is that it indicates lower memory consumption or fewer flops. Is that the case when comparing the baseline ResNet with the proposed CapsNet? Otherwise, a set of experiments showcasing the viewpoint generalizability of proposed CapsuleNetworks might only justify the switch between resnets to the proposed capsnets.

Thirdly, Fig. 4 top images seems to indicate all 3 routing procedures are following the same Learning Rate schedule. In the text it is said that optimization hyperparameters are tuned individually. Did authors tune learning rate schedule individually?

Forth, the proper baseline for the current study is the dynamic routing CapsNet. Why the multiMNIST experiment lacks comparison with dynamic routing capsnet?

For the reasons above, the manuscript in its current format is not ready for publication.

------------------------------------------------------rebuttal
Thank you for your response. I acknowledged the novel contributions of this work. My comment was that some claims in the paper are not right. i.e. "inverted dot-product attention" is not new and "reconstructing the layer bellow" does not happen in Sabour et al . Parallel execution + layer norm definitely is novel and significant.

Regarding the LR-schedule, I am not sure how fair it is to use same hyper-params tuned for the proposed method on the baselines.

Regarding the viewpoint, the diverseMultiMNIST is two over lapping MNIST digits shifted 6 pixels. There is no rotation or scale in this dataset. An example experiment verifying the viewpoint generalizability of the proposed model is training on MNIST testing on AFFNIST.


**Experience Assessment:**

I have published in this field for several years.

**Review Assessment: Checking Correctness Of Derivations And Theory:**

I carefully checked the derivations and theory.

**Review Assessment: Checking Correctness Of Experiments:**

I carefully checked the experiments.

**Review Assessment: Thoroughness In Paper Reading:**

I read the paper thoroughly.

---

> ### Author Response · Authors · 2019-11-14
> **Reponse to Review #4**
>
> We thank the Reviewer for constructive feedback. We hope the following response will address the concerns of the Reviewer.
>
> [Remarks on Inverted Dot-Product Attention Routing]
> We agree that our routing method has similar components to Dynamic Routing (Sabour et al 2017), and we would like to emphasize their differences : 1) Sequential iterative routing is replaced with concurrent iterative routing, 2) Squash activation is replaced with Layer Normalization, and 3) We use cross-entropy loss instead of margin loss. The comparison is summarized in Section 4.3.
>
> We humbly argue that these modifications are not trivial and stabilize the training, which leads to improved performance. For example, we observe that only our model has improved performance when the routing iteration number increases (CIFAR10 classification Table of Figure 4).
> ============================================
> Method   | Iteration=1 | Iteration = 3 | Iteration = 5
> Dynamic  |    84.08%     |    82.88%       |    82.11%
> EM            |    58.08%     |    78.43%       |    31.41%
> Ours         |    84.24%     |    84.83%       |    85.09%
> ============================================
>
> [Remarks on using ResNet]
> We agree that using a deeper CNN such as a ResNet (vs. a single convolutional layer) to produce primary capsules makes our model inherit the disadvantages of CNNs (such as less view-point generalizability) and blunts the potential impact of capsules. However, at this stage, our intent is not to replace CNNs completely with CapsNets, but take a meaningful step towards building a routing mechanism that can at least do the job of the higher layers of a CNN. Previously proposed routing algorithms fail to do so and perform worse than their baseline CNNs.
> ==================================================
> Method                                                                      |  Accuracy
> -----------------------------------------------------------------------------
> Dynamic routing with DenseNet backbone [1]  |   89.71%
> -----------------------------------------------------------------------------
> Dynamic routing with ResNet backbone             |   92.65%
> EM routing with ResNet backbone                       |   92.15%
> Our routing with ResNet backbone                      |   95.14%
> -----------------------------------------------------------------------------
> Original ResNet                                                        |   95.11%
> ==================================================
> [1] Phaye et al. “Dense and Diverse Capsule Networks: Making the Capsules Learn Better.” Arxiv 2018.05.
>
> [Remarks on Memory Consumption]
>
> Ｗe agree with the Reviewer that reporting only the number of parameters may not be satisfying. Therefore, in Figure 5, we report the memory consumption comparisons between CapsNets and CNNs given the same model architecture. Please see the response to Reviewer #1, where we outline the reasons for why CapsNets consume more memory compared to CNNs even with fewer parameters, and we also suggest some possible solutions on reducing memory consumption, which we leave as our future work.
>
> We also like to point out that the networks with fewer model parameters but larger runtime memory footprint may still be preferable for certain IC designs, where the L1 cache can store all the parameters.
>
> [Learning Rate Scheduler]
> We use the same learning rate scheduler for all three models. The learning rate degrades by 0.1 on the 150th and 250th epochs. In the original submission, we have included these details in Section A.1 in Supplementary.
>
> [Dynamic Routing Methods for DiverseMultiMNIST]
> We provide the results for the Dynamic routing method by applying it on the DiverseMultiMNIST dataset. For a fair comparison, we consider the same optimizer, the same number of layers, and the same number of neurons per layer for the Dynamic routing method and the other methods.
>
> The results are highlighted below (CapsNet denotes our routing method):
> ================================================
> Method    |  Pose Structure  | Test Acc.  |   # params.
> -----------------------------------------------------------------------------
> Dynamic  |      vector              |  83.39%    |   42.48M
> -----------------------------------------------------------------------------
> CapsNet   |      matrix             |  80.59%    |   9.96M
> CapsNet   |      vector              |  85.74%    |   42.48M
> -----------------------------------------------------------------------------
>             BaselineCNN                |  79.81%    |   19.55M
> ================================================
>
> Compared to the Baseline CNN, both our routing method and the Dynamic routing method achieve better performance on the DiverseMultiMNIST dataset. This result suggests a better viewpoint generalization from CNNs to the Capsule networks. Furthermore, our routing method outperforms the Dynamic routing one. We have updated our manuscript with these results.

---

### Public Comment · ~Hopeful_Rational2 · 2019-10-04
**Request for code**

Hi.
I feel that this is a good step forward getting capsules closer to state-of-the-art on complicated datasets like CIFAR10.
Could you please release the code ASAP.
Thanks.

---

> ### Author Response · Authors · 2019-10-05
> **Response to Code Release**
>
> We are cleaning the code, and planning to release it once it is ready.

---

### Author Response · Authors · 2019-11-14
**Manuscript Revision**

We have updated the manuscript, and we highlight the additional results/ discussions in red.

---

### Decision · Program_Chairs · 2019-12-19

**Decision:**

Accept (Poster)

**Comment:**

This work presents a routing algorithm for capsule networks, and demonstrates empirical evaluation on CIFAR-10 and CIFAR-100. The results outperform existing capsule networks and are at-par with CNNs. Reviewers appreciated the novelty, introducing a new simpler routing mechanism, and achieving good performance on real world datasets. In particular, removing the squash function and experimenting with concurrent routing was highlighted as significant progress. There were some concerns (e.g. claiming novelty for inverted dot-product attention) and clarification questions (e.g. same learning rate schedule for all models). The authors provided a response and revised the submission , which addresses most of these concerns. At the end, majority of reviewers recommended accept. Alongside with them, I acknowledge the novelty of using layer norm and parallel execution, and recommend accept.